# Educational Intervention in Rehabilitation to Improve Functional Capacity after Hip Arthroplasty: A Scoping Review

**DOI:** 10.3390/jpm12050656

**Published:** 2022-04-19

**Authors:** Célia Nicolau, Liliana Mendes, Mário Ciríaco, Bruno Ferreira, Cristina Lavareda Baixinho, César Fonseca, Rogério Ferreira, Luís Sousa

**Affiliations:** 1Alentejo Coastal Hospital (HLA), Alentejo Coastal Local Health Unit (ULSLA), 7540-230 Santiago do Cacém, Portugal; mendesliliana034@gmail.com (L.M.); mario.paz.ciriaco@gmail.com (M.C.); 2Nursing Research, Innovation and Development Centre of Lisbon (CIDNUR), 1900-160 Lisbon, Portugal; brunoamferreira@gmail.com (B.F.); crbaixinho@esel.pt (C.L.B.); luismmsousa@gmail.com (L.S.); 3Beatriz Angelo Hospital, 2674-514 Loures, Portugal; 4Nursing School of Lisbon, 1900-160 Lisbon, Portugal; 5Comprehensive Health Research Centre (CHRC), 7000-811 Évora, Portugal; cfonseca@uevora.pt (C.F.); ferrinho.ferreira@ipbeja.pt (R.F.); 6Nursing Department, University of Évora, 7004-516 Évora, Portugal; 7Polytechnic Institute of Beja, 7800-111 Beja, Portugal; 8Escola Superior de Saúde Atlântica, 2730-036 Lisbon, Portugal

**Keywords:** hip arthroplasty, nursing care, rehabilitation, patient education, mobilisation, patient-centred care

## Abstract

(1) Background: The prevalence of coxarthrosis associated with ageing is one of the main reasons for the increase in hip arthroplasty. Educational intervention in the pre- and postoperative period of hip arthroplasty can improve functionality and, consequently, quality of life. Therefore, we intend to map the educational interventions that improve the functional capacity of people undergoing hip arthroplasty on their rehabilitation process. (2) Methods: Scoping review. The search was carried out in the databases: CINAHL complete, MEDLINE Complete, Nursing and Allied Health Collection: Comprehensive, Cochrane Methodology Register, Library, Information Science and Technology Abstracts, MedcLatina, and Google Academic. We included studies that evaluated the implementation of educational rehabilitation interventions that enable the empowerment and teaching of people undergoing hip arthroplasty. (3) Results: Thirteen studies were analysed that met the eligibility criteria and allowed the research question to be answered. Education in the perioperative period promotes training to perform rehabilitation exercises, improves the hospitalisation experience, increases participation in the rehabilitation process, allows early mobilisation, and increases the functional potential, thus enabling faster reintegration into society. (4) Conclusion: Early mobilisation plays a very important role in a person′s functional recovery, as well as in the prevention of thromboembolic complications. Exercise produces benefits in relation to physical function, namely, in improving strength, balance, facilitating functional activities, and preventing complications.

## 1. Introduction

The ageing process, with its numerous anatomical and physiological changes, contributes to the increased prevalence of chronic diseases and the consequent increase of people in a situation of dependency [1,2]. These physiological changes manifest themselves with balance deficits; postural control maintenance; muscle atrophy; weakness; and changes in cognitive functions such as memory, learning, and awareness, with contributions to the increased risk of falling and the decreased ability to perform activities of daily living (ADLs) [1].

With aging, joints suffer progressive wear and tear of the cartilage and structures involved, with osteoarthritis being the leading cause of functional disability and the hip joint being one of the most affected [1,2]. These factors, together with an increase in average life expectancy, mean that a large part of the elderly population suffers from pain and limited functional capacity to perform ADLs [2]. In addition, we must also consider the loss of bone and muscle mass, which leads to a greater predisposition to falls and consequently fractures, with the most frequent ones occurring in the upper extremity of the femur, namely, in its neck and trochanteric region. This increases considerably after the age of 60, with a maximum incidence between the ages of 70 and 79 [3].

These phenomena have a great impact on the elderly population, contributing to a significant increase in morbidity and mortality, as well as to an increase in the likelihood of long-term institutionalisation due to dependence, which implies high social and economic burdens [3].

Hip arthroplasty emerges as a solution when there is no improvement of the symptoms with conservative treatment. This surgical procedure is widely used with great success in the treatment of joint disease, whether degenerative, inflammatory, or traumatic, promoting pain reduction, recovery of limb function, reduced disability, and improvement of life quality [4].

A structured intervention programme developed by rehabilitation nurses, with the aim of empowering people to achieve the greatest possible autonomy and independence in self-care, plays a key role, as the need for functional recovery after surgery is extremely important, both in terms of restoring functional capacity and returning to social and professional life [5]. In the preoperative period, the intervention of the rehabilitation nurse should focus on education as a strategy to prevent complications, providing information about the surgical procedure and rehabilitation exercises with the goal of promoting functional adaptation to ADLs. Respiratory functional re-education should also be addressed, with the aim of preventing respiratory complications [3].

Within the scope of functional motor re-education, the following should be addressed: positioning in bed; isometric contractions of abdominal, gluteal, and quadricep muscle groups; lumbopelvic extension or half-bridge; getting up and lying down in bed; sitting down and getting up from a chair; using the shower/bath; using the toilet; walking with a Zimmer frame or crutches up and down the stairs; getting in and out of the car; mobilisation of the tibiotarsal joint to facilitate venous return and prevent complications such as thrombophlebitis; and also the movements to be avoided in order to prevent dislocations [3,6].

Various studies have shown that health education interventions contribute towards an increase in the satisfaction of the person and their family, as well as decreasing the administration of painkillers, reducing the risk of dislocation, helping with rehabilitation, and facilitating functional activity. According to some authors, these interventions can improve self-efficacy to regain health and facilitate rehabilitation after surgery [7,8,9]. One study, which compared a standardised care approach to the person-centred care for patients undergoing total hip replacement surgery, concluded that 3 months after surgery, 88% of patients in the control group had regained their independence versus 92.5% in the person-centred care group [9].

This emphasises the importance of focusing our attention on people and their families, making them partners in the decision-making process about the care to be provided [8,9], as co-responsibility increases adherence to the rehabilitation programme [8]. The first step towards this is information and communication strategies adjusted to the literacy level of the person and their family [8,9].

This review aims to present the educational interventions that improve the functional capacity of people undergoing hip arthroplasty on their rehabilitation process.

## 2. Materials and Methods

### 2.1. Study Design

Considering the defined objective and given the heterogeneity of the studies interventions, we chose to conduct a scoping review, as it is a comprehensive form of research used to address broad topics, focusing on in-depth results based on the available evidence [10,11,12], which allow us to chart the educational interventions developed in the rehabilitation of people who have undergone hip arthroplasty, as well as their outcomes.

On the other hand, in addition to identifying, this method allows us to examine and systematise a concept or its characteristics by allowing the identification of the nature of a broad field of knowledge [10].

A protocol (OSF Registration: osf.io/7362b) was followed to answer the research question: “What are the rehabilitation interventions performed to empower and enable the person undergoing hip arthroplasty to improve functional capacity?”, with 6 steps: (1) Identification of the review question using the acronym PCC as a starting point; (2) Designation of the inclusion and exclusion criteria of studies and identification of the relevant ones; (3) Selection of the studies; (4) Assessment of the level of evidence of the collected literature, according to JBI guidelines; (5) Discussion of the results; and (6) Synthesis and presentation of the results obtained [13].

### 2.2. Elegibility Criteria

As inclusion criteria of the review, the following were established:
P—(Population) adults and older persons’ who underwent hip arthroplasty;C—(Concept) educational rehabilitation interventions;C—(Context) hospital and rehabilitation units.

Studies exploring educational programmes in people with various types of arthroplasty were accepted, provided that they allowed the extraction of data concerning the sample of people with hip arthroplasty. Systematic reviews and quantitative primary studies were included. The exclusion criteria considered were articles with children and teenagers, unrelated to the topic under study, with ambiguous methodology and repeated in both databases.

### 2.3. Data Collection

The search descriptors were: “hip arthroplasty”, “Nursing care”, “Rehabilitation”, “Patient education”, and “mobilization” validated in MeSH and DeCS. The Booleans, “OR” and “AND” were used as well as the expander apply to equivalent subjects.

The search was performed on the EBSCOhost platforms (MEDLINE, CINAHL; Cochrane; MedicLatina) and Google Academic, during the month of April 2021, for studies in Portuguese, English, French, and Spanish (Table 1).

Study selection involved title evaluation and abstract analysis in order to identify whether the articles met the inclusion and exclusion criteria. Two reviewers independently assessed the inclusion of studies by reading the titles, abstracts, and keywords, excluding those that did not meet the inclusion criteria for this review. Subsequently, the same reviewers independently assessed the full texts. In cases of discrepancies in the decision, a third reviewer was used. The articles were also assessed by two independent reviewers for the level of evidence before inclusion in the review. A third reviewer was also included in divergent and doubtful cases.

### 2.4. Data Processing and Analysis

In the data extraction phase, a descriptive analysis of each study was performed using an extraction instrument designed to extricate the information according to the research question. The isolated data had specific details about the study objective, study design, assessment instruments, participants, interventions, and main findings.

Data extraction was carried out by two reviewers independently, and doubts and disparities were resolved with the inclusion of other two reviewers.

In the data extraction phase, the content of the articles was thoroughly analysed, which allowed not only for the answering of the research question, but also for the observation of the contribution of educational interventions in the rehabilitation programmes of patients who had undergone hip arthroplasty, with regards to pain control, reduction of complications, hospitalisations, readmissions, and recovery time.

## 3. Results

In the EBSCO Host database, 546 studies were identified. After removal of the duplicated results, only 462 were left for analysis. Of these 462 studies that were identified, 416 were excluded by evaluating the title, leaving 46 for further reading.

After reading the abstract, a further 25 studies were excluded as they did not answer the research question. The full-text versions of the remaining 21 articles were read, and seven articles met the inclusion criteria.

In Google academic, 140 studies were identified, and after the analysis of the titles, only 10 were left for further analysis. After the full reading of the studies, six were selected to be part of this review; so, in total, 13 studies were included.

The selection process is presented through the PRISMA-ScR flowchart with the results in the different phases.

The identification and selection of studies for inclusion are presented in Figure 1.

The 13 studies included in the present scoping review were published between 2003 and 2021, with one of the articles published in 2003 [14], one in 2005 [7], one in 2008 [15], two in 2011 [16,17], one in 2012 [18], one in 2018 [19], one in 2019 [20], and five in 2020 [21,22,23,24,25].

Eleven articles are in English, although with various countries of origin, such as Canada [17], Sweden [14], Taiwan [7], Germany [16], Italy [19], the United Kingdom [18], China [21,23], Thailand [20], Denmark [22], the United States of America [24], and Portugal [25]. The French article originated in France [15], and the Portuguese article originated in Portugal [25] (Table 2).

Of these articles, 11 are primary sources: cohort study [16,17,20], quasi-experimental study [7,18,19], randomised controlled study [21,22,23,24]; two are systematic reviews [14,15]; and one is a quantitative descriptive study [25].

The number of participants ranged from 12 [19] to 242, divided into control group (124) and experimental group (118) [24].

The main interventions were carried out at the level of preoperative education through visualisation of leaflets, videos, and CDs; an effective postoperative analgesia; and early mobilisation. This includes isometric and isotonic mobilisations of the lower limbs, transfers to bed and chair, and gait training with gait assistants.

Preoperative education contributes to the satisfaction of the patients with the nursing care provided and reduces anxiety of the unknown, since all the patient’s doubts are clarified, preparing them for surgery and rehabilitation. Early mobilisation is beneficial in terms of functional mobility, contributing to the reduction of hospital length of stay and the incidence of postoperative complications, while promoting an improvement in the quality of life of hip arthroplasty patients and facilitating their reintegration into the family and society.

After assessing the levels of evidence (Table 2), the results were analysed. The extracted data included specific details on the study objective, study design, assessment tools, participants, interventions, and main conclusions (Table 3).

The analysis of the studies in the literature sample showed that educational interventions are maintained throughout the perioperative period, focusing on information about the surgery and changes in self-care and the teaching of early mobilisation, transfer, gait, and safe ADL training to avoid luxating movements. Some programmes also assist in managing peri-operative expectations and hospitalisation to make the experience less traumatic.

## 4. Discussion

The 13 studies included in this scoping review allowed us to chart the educational interventions that have the potential to improve the functional capacity of people undergoing hip arthroplasty on their rehabilitation process. In addition, it was possible to identify secondary beneficial outcomes such as improvement of pain control; reduction of complications; and decreased hospital stay, readmissions and recovery time.

A preoperative rehabilitation programme that includes a home visit allows for a correct identification on the required support devices, adaptive equipment, and the person’s perception of their abilities. It makes the educational intervention more effective and allows for the ADL training to be performed as close to reality as possible [25].

It is through nursing interactions, such as teaching and guidance, that there is an increase in the person’s knowledge and empowerment, teaching them to make decisions. Learning skills allows the development of effective strategies in the recovery of stability and wellbeing, permitting the maximisation of autonomy and facilitating the performance of basic and instrumental activities of daily living. By doing so, there is an improvement in the participation in society and in people’s quality of life [27].

Educational interventions were carried out in sessions and the themes usually approached were the surgery, the risks and complications, breathing exercises before surgery, isometric exercises of the quadriceps, dorsiflexion and plantar flexion of the ankle, postoperative care, thromboembolic prophylaxis, first standing up, gait training with the aid of devices, and care to prevent dislocation of the prosthesis [7,16,17,18,19,23,25].

These interventions can be reinforced with the support of paper instructions [7,17,20,22,25], viewing of videos and CDs [7,25], and monitoring of physical activity and pain by text messages [24], which contributes to better self-efficacy and increased functional capacity. Monitoring has a more significant effect on reducing the readmission rate by providing a direct mean of communication with the health professional to clarify doubts and assess urgent problems [24].

Other studies argue that eHealth programmes that support individualised education of the person in preoperative preparation, inpatient care, and home rehabilitation have the advantage of potentially increasing the person’s involvement, improving their recovery, and reducing potential postoperative complications [28,29], contributing to positive clinical outcomes, when compared to conventional face-to-face rehabilitation approaches. These are low-cost interventions suited to their specific needs and with low levels of disruption in their daily life, which could justify the implementation of telerehabilitation in clinical settings, not only in the current COVID-19 context but also in post-surgery follow-ups [30]; in addition, these interventions have very significant effects on reducing hospitalisation time, without altering the rates of complications and readmissions [31].

The articles in the literature sample indicate that educational interventions may also help with pain control [16,18,23], which corroborates the results of other studies that confirm an approach focused on educational interventions, patient-centred care, and their fears and anxieties contribute to an effective pain control [8,9,32]. After surgery, effective pain control is one of the most important issues, and it is necessary to optimise pain control, whether through preoperative analgesia as a preventive measure [16,17,18,23], infiltration of local anaesthetic into the wound [18], intra-articular injection of local anaesthetic [16,17], or the use of self-controlled intravenous analgesia (PCA) [17]. All these alternatives had a significant influence on adequate pain control and the possibility of mobilisation on the day of surgery.

Effective pain management and educational intervention performed by nurses may help early recovery after hip arthroplasty. Therefore, it is essential that nurses ensure that people receive high quality and effective education. The development of effective education interventions has been described as one of many ways to ensure quality nursing in the future [33].

Early mobilisation has been shown to be effective in restoring function and mobilisation, relieving pain, and speeding gait recovery, which, whenever possible, should be initiated during the first four hours after surgery [16,17,18].

Another measure to promote early mobilisation and standing up is the preference of administration of tranexamic acid [16] over the placement of drains [16,18] as a measure to reduce blood loss. This option favours early mobilisation and standing up.

A preoperative functional exercise plan improves the person’s ability to adjust to the surgery, and postoperative functional exercises promotes the rehabilitation of limb function, whilst significantly reducing hospitalisation time and the incidence of postoperative complications, as well as improving the people′s quality of life [15,19,21]. The educational intervention on physical exercise should be tailored to on the needs and expectations of people undergoing total hip arthroplasty as this shows benefits in reducing levels of pain and improving activities of daily living [34].

Preoperative resistance training has not shown additional benefit regarding muscle strength, but it is a viable complement to achieve the earliest onset of postoperative functional recovery, as it has been shown to accelerate rehabilitation by three months [22].

The implementation of rapid recovery programmes after hip arthroplasty can be very beneficial for the person in terms of pain control and recovery of mobility, as well as in reducing the length of hospital stay, which brings benefits in terms of hospital costs [14,16,17,18,23]. Several studies have shown that enhanced recovery after surgery (ERAS) reduces the need for prolonged hospital stays, the reintervention rate, and the readmission rate, with consequent savings in hospital costs [35,36]. These types of programmes, based on interventions to reduce psychological trauma and postoperative stress, accelerate the recovery process and contribute significantly to increasing patient satisfaction, in addition to the aforementioned effects [35,37].

Given the type of studies included in this study are more of a quantitative paradigm, we believe it is important to conduct further qualitative research to understand not only people’s satisfaction with the programmes, but also to assess which interventions and educational strategies are better suited to the literacy of this population, as that will allow for better clinical decision making using the person’s preferences, experience, clinical context, and resources [38].

The above is in line with the consensus guidelines on the subject: preoperative education and counselling is strongly recommended as it reduces anxiety; physiological optimising the person in pre-admission, by decreasing existing preoperative risk factors, such as smoking, alcohol, anaemia, nutritional and metabolic status, and low physical activity, is also strongly recommended as it contributes to the reduction of complications and consequently the reduction of hospitalisation days; the prevention of perioperative blood loss through the use of tranexamic acid is strongly recommended, thus reducing the need for blood transfusions; the use of effective postoperative analgesia; the maintenance of normothermia during the intraoperative period; the use of antimicrobial and antithrombotic prophylaxis; early mobilisation, where people should be mobilised as early as possible in order to prevent risks associated with immobilisation and criteria-based discharge. The requirements for discharge from the hospital include the ability to dress independently, to get in and out of bed, to sit up and get out of a chair or toilet, and the ability to independently care for their personal hygiene and mobilise with a walking aid [39].

A home rehabilitation programme is considered safe and effective in improving the physical performance and quality of life of people who have undergone hip arthroplasty [20]. Early mobilisation is the gold standard for recovering functional mobility, as exercises can produce long-term benefits in relation to physical function, namely, improving strength and balance, facilitating the performance of functional activities, and preventing adverse outcomes such as falls.

Postoperative rehabilitation programmes in the community consist of functional re-establishment and the use of educational approaches, allowing gains in functionality greater than conventional rehabilitation strategies [40].

The transition from hospital care to outpatient care or home settings requires significant paradigm changes, which is not widely used and is mainly implemented in outpatient surgery centres or specialised orthopaedic hospitals, despite the reduction of complications and readmissions [41], with high levels of satisfaction for the person and the caregiver, as it allows for the clarification of the process and expectations [8,42]. There is evidence of individualised transition care programmes implemented and led by nurses that show gains in functionality in the postoperative period of hip arthroplasty [43].

The transition from hospital to home should be made safely, with the certainty that the person and their family will receive the necessary support according to their needs. The nurse’s role is important in defining appropriate interventions to ensure continuity of care between hospital and home, minimising side effects and secondary disabilities caused by the surgery whilst promoting the knowledge and capacities of the person and their family in adapting to their new reality. The nurse has the responsibility to ensure that discharge occurs safely and people and their families feel properly prepared and supported [44,45], which highlights the need to involve the family in this preoperative process, enabling them to provide continuity of care, while promoting the independence of the person undergoing surgery.

### Limitations

One limitation is related to the very design of a scoping review, as it aims to provide a view of the state of research in relation to the questions posed, and the quality of the included studies is not considered. Another limitation is the inclusion of studies only in Portuguese, French, Spanish, and English.

## 5. Conclusions

From the studies reviewed, we can determine that rehabilitation interventions play a key role in the functional recovery after hip arthroplasty. The empowerment of the person/family in relation to the process they will experience allows for a more positive vision, less stress, and greater participation.

A programme that includes a preoperative assessment at the person’s home allows for the correct identification of the support devices required, the person’s perception of their abilities, and the characteristics of the home itself, so that the training of life activities can be carried out as close as possible to reality.

The use of education strategies alone does not reduce pain, functioning, or hospitalisation time, so adequate pain management is essential, with effective administration of analgesics, so that the person can start mobilisation as early as possible. This early mobilisation plays a very important role in the person’s functional recovery, as well as in the prevention of thromboembolic complications. Exercise benefits physical function, improving strength and balance, assisting functional activities, and preventing complications.

Teaching should combine verbal and written resources and images and should be exemplified whenever possible. The follow-up after discharge is very important, allowing for counselling and clarification of doubts and avoiding readmissions. In the current pandemic context, and with the technological resources that most people have access to, teleconsultation may be used as a strategy for follow-up in the postoperative period or even in the preoperative consultation. This topic should also be considered for further studies.

In conclusion, all of these are vital aspects to be considered and implemented, as they allow patient empowerment, increasing their functional potential and facilitating their reintegration into society.

## Figures and Tables

**Figure 1 jpm-12-00656-f001:**
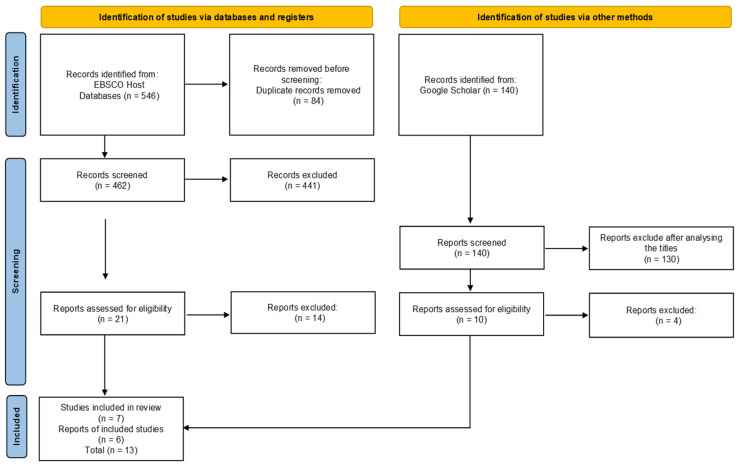
PRISMA–ScR flowchart. Lisbon, 2020.

**Table 1 jpm-12-00656-t001:** Syntaxes of combined descriptors in the scientific database search. Lisbon, 2020.

Database	Syntax Adopted
Medline	(((((((hip arthroplasty[Title/Abstract]) OR (hip replace*[Title/Abstract])) OR (hip[Title/Abstract])) OR (arthroplasty[Title/Abstract])) OR (surger*[Title/Abstract])) OR (arthroplasties, hip replacement[MeSH Terms])) AND ((((((((educational rehabilitation interventions[Title/Abstract]) OR (educ*[Title/Abstract])) OR (reab*[Title/Abstract])) OR (intervent*[Title/Abstract])) OR (activities, educational[MeSH Terms])) OR (health education[MeSH Terms])) OR (care, self rehabilitation[MeSH Terms])) OR (early intervention education[MeSH Terms]))) AND (((((hospital[Title/Abstract]) OR (rehabilitation units[Title/Abstract]))) OR (center, rehabilitation[MeSH Terms])) OR (hospitalization[MeSH Terms]))
CINAHL	(TI (“hip arthroplasty” or hip*) and (“Nursing care” or “Rehabilitation” or “Nurs*” and (“Patient education” or “mobilization”) (TI hip arthroplasty OR hip replace OR hip OR arthroplasty OR surger*) OR (AB hip arthroplasty OR hip replace OR hip OR arthroplasty OR surger*) OR (MM “Arthroplasty, Replacement, Hip”)) OR (MM “Hip Surgery”) AND (TI educational rehabilitation interventions OR educ* OR intervent*) OR (AB educational rehabilitation interventions OR educ* OR intervent*) OR ((MH “Nursing Interventions”) OR (MM “Outcomes of Education”) OR (MM “Rehabilitation, Geriatric”)) OR (MM “Nursing Care”) OR (MM “Patient Education”)) AND (TI hospital OR rehabilitation units) OR (AB hospital OR rehabilitation units) OR (MH “Hospital Programs”)) OR (MM “Rehabilitation”)).
Cochrane	(“hip arthroplasty” OR hip*) AND (“Nursing care” OR “Rehabilitation” OR “Nurs*” OR “Patient education” OR “mobilization”) AND (Hospital OR Rehabilitation unit)
MedicLatina	(“hip arthroplasty” OR hip*) AND (“Nursing care” OR “Rehabilitation” OR “Nurs*” AND (“Patient education” OR “mobilization”) AND (Hospital OR Rehabilitation unit)
Google academic	(“hip arthroplasty” OR hip*) AND (“Nursing care” OR “Rehabilitation” OR “Nurs*” OR “Patient education” OR “mobilization”) AND (Hospital OR Rehabilitation unit)

**Table 2 jpm-12-00656-t002:** Classification of included articles by level of evidence and country of origin. Lisbon, 2020.

Reference	Level of Evidence [26]	Country
[7]	2.c—Quasi-experimental study	Taiwan
[14]	2.b—Systematic review of quasi-experimental studies	Sweden
[15]	2.b—Systematic review of quasi-experimental studies	France
[16]	3.c—Retrospective cohort study	Germany
[17]	3.c—Cohort study	Canada
[18]	2.c—Quasi-experimental study	United Kingdom
[19]	2.c—Quasi-experimental study	Italy
[20]	Prospective cohort study	Thailand
[21]	1.c—Randomised controlled study	China
[22]	1.c—Randomised controlled study	Denmark
[23]	1.c—Randomised controlled study	China
[24]	1.c—Randomised controlled study	USA
[25]	4—Descriptive, retrospective study of a quantitative nature	Portugal

**Table 3 jpm-12-00656-t003:** Data extraction. Lisbon, 2020.

Study/Sample	Study Design and Aim	Intervention	Results
[7] 33 people in the control group and 33 people in the experimental group	Quasi-experimental study. To examine the effects of multimedia with printed nursing guides in the education of the person with hip arthroplasty, on the improvement of self-efficacy, functional activity, and length of hospital stay.	The control group received standard care, which included individual education with leaflets during hospitalisation. The experimental group received a nursing guide in CD and printed forms, with video and audio nursing instructions that addressed aspects about articulation, preparation for surgery, use of assistive devices, and rehabilitation.	The results showed that the experimental group, which received education through multimedia CD and printed nursing guides, showed statistically better self-efficacy and functional activity, and shorter hospital stay than the control group.
[14]	Systematic review. To know the exercise-based treatments applied during the postoperative period and the possible implications for discharge destination and health outcomes.	Instruction for early mobilisation is the gold standard for achieving functional mobility, including sufficient range of motion. Exercise improves physical activity-related outcomes after hip arthroplasty.	Patients can achieve similar pain relief and functional capacity when discharged home with therapeutic exercise supervision compared to discharge to a rehabilitation facility. Exercise produces long-term benefits regarding physical function in elderly, improving strength, balance, and other neuromuscular aspects, facilitating functional activities and potentially preventing adverse outcomes such as falls.
[15] 200 expert professionals	Systematic review. Developing clinical practice guidelines for early mobilisation following total hip arthroplasty.	Education addresses early mobilisation, transfer, walking, and prevention of dislocating movements. Early mobilisation after hip arthroplasty is an integral part of post-operative management, in the approach to the person and the operated hip.	This early mobilisation plays a crucial role in the person’s initial functional mobilisation, as well as an important role in the prevention of thromboembolic diseases. After this initial period (3 to 4 days), the attention is focused on the prevention of movements with risk of dislocation, the functional transference, and the beginning of gait.
[16] 102 people submitted to the rapid treatment programme in hip arthroplasty	Retrospective cohort study. Assessment of function, mobilisation, and pain scores during hospitalisation (6 postoperative days) and 4 weeks after fast-track treatment in hip arthroplasty.	Conducting a multidisciplinary lecture preoperatively and gait training with crutches for all persons, administration of preventive non-steroidal anti-inflammatory drug one hour before the intervention, minimally invasive anterolateral approach under spinal anaesthesia, administration of intravenous dexamethasone, placement of uncemented implants, infiltration of analgesia in the peri-acetabular and femoral region, and administration of tranexamic acid.	The application of a fast-track programme was effective regarding function and mobilisation, as well as pain relief and gait recovery speed.
[17] 100 people treated in a fast-track programme compared with 100 people treated before the introduction of this programme	Cohort study. Primary aim: to determine whether a fast-track care model can reduce the length of hospital stay following hip and knee arthroplasty while maintaining the person’s safety. Secondary aim: to compare the incidence of clinically significant outcomes of the fast-track programme with the previous common care programme.	The fast-track programme emphasises pre- and postoperative education, postoperative analgesia with periarticular injection, early mobilisation, and discharge home with an outpatient rehabilitation programme. All were contacted by the nurse 2 to 3 days after discharge for symptom assessment and recovery.	The fast-track programme can reduce postoperative length of stay while maintaining appropriate pain management and safeguarding the person′s satisfaction and safety. All people reported a good surgical and hospital experience.
[18] The first 95 consecutive people who underwent total hip arthroplasty or total knee arthroplasty	Quasi-experimental study. To report the outcomes of rehabilitation following a Norwich Enhanced Recovery Programme (NERP) in terms of function and pain at discharge, length of stay, need for rehabilitation services after discharge, and complications during the first 6 weeks after total hip and knee arthroplasty.	All participated in a preoperative educational session that included information on the route within the hospital, postoperative exercises, and advice on physical abilities expected postoperatively; people who were to undergo THA were also taught about care to prevent prosthesis dislocation. In THA surgeries, all implants were cemented, local anaesthetic was injected into the operative wound, and a catheter was placed in the wound for infiltration of local anaesthetic on the first 12 h; no drains were used. Approximately 4 h after surgery, with the local anaesthetic still effective, an assessment was made with the intention of starting the exercises; to do the lifting; and, with a gait aid, to do gait training.	The results of this study indicate that the development of the Norwich Enhanced Recovery Programme (NERP) was a successful rehabilitation regime for patients undergoing total hip arthroplasty (THA) and total knee arthroplasty (TKA), facilitating early and safe discharge with minimal complications. This suggests that initiating mobilisation within 4 h of surgery was important in improving initial functionality outcomes, as well as reducing pain levels and length of stay.
[19] 12 consecutive people submitted to total hip arthroplasty	Quasi-experimental study. To evaluate the feasibility and effectiveness of an intensive rehabilitation programme after hip arthroplasty in an Italian spa centre.	All people underwent a 2-week thermo-multimodal rehabilitation programme, which consisted of educational and physical rehabilitation measures. The rehabilitation treatment consisted of six sessions/week of rehabilitation, gait training and balance strategies, kinesiotherapy, and hydro kinesiotherapy in thermal pool. The educational programme was conducted for the people and their families.	The study showed that this intensive treatment was feasible in a thermal spa and was effective, producing good results in terms of pain relief, improving motor and functional capacity, and improving people′s perception of quality of life. Thermal centres with a vocation for rehabilitation can provide various types of rehabilitation procedures, such as physical therapies (electrical, ultrasound, among others) and various forms of kinesiotherapy and functional training, such as passive/active mobilisation, hydro kinesiotherapy, respiratory training, balance and gait training, and health prevention programmes, in addition to traditional thermal therapy.
[20] 41 people submitted to cementless bipolar hemiarthroplasty after femoral neck fracture	Prospective cohort study. To investigate the effectiveness of a home-based rehabilitation programme, examining recovery time, risk of falling, improvement in mobility, and improvement in quality of life.	After surgery, participants received instructions on how to perform a home rehabilitation programme. They had to perform exercises every day for 6 months, 10–15 repetitions, 2 sets/3 × a day, including lying down and standing position exercises. The lying down exercises consisted of hip abduction and hip flexion. The standing exercises included hip abduction, extension, and flexion.	The home rehabilitation programme in this study was found to be safe and effective in improving the recovery of people undergoing hip hemiarthroplasty in physical performance and quality of life. All participants were able to return to their pre-injury status within six months.
[21] 150 patients who were divided into a control group (75) and an experimental group (75)	Randomised controlled trial. To explore the effect of a functional exercise nursing plan in patients following hip replacement.	The control group received conventional rehabilitation treatment. The experimental group underwent a specific functional exercise nursing plan based on relevant evidence, studies, and methods from the literature, combined with the person′s particular situation and guided by nursing.	The results obtained show that a functional exercise nursing plan based on existing literature and studies in people after hip arthroplasty has a significant effect. It promotes their recovery, improves their quality of life, and reduces the length of hospital stay and the incidence of postoperative complications.
[22] 80 patients who were divided into a control group (40) and an experimental group (40)	Randomised controlled study. To investigate the postoperative effect at 12 months of preoperative resistance training in patients undergoing hip arthroplasty on activity and function and expected outcomes on muscle strength and physical performance.	The experimental group participated in a preoperative training program, performed in sessions of 1 h, 2 times a week for 10 weeks. Each session included 10 m warm-up followed by a sequence of 4 exercises performed on training machines (hip extension, knee extension/flexion, and leg press); the exercises were performed at moderate to high intensity in 3 sets of 8 to 12 repetitions with a load adjusted to each person, not causing any pain. The control group received exercises to perform at home, of low intensity without specific resistance exercises.	The results obtained showed that after 12 months of surgery, the preoperative resistance training programme did not provide additional benefit regarding muscle strength and additional tests; however, it provided accelerated rehabilitation by 3 months. Therefore, intensive preoperative training is a viable complement to achieve the earliest onset of postoperative functional recovery after hip arthroplasty.
[23] 58 patients divided into a control group (29) and an experimental group (29)	Randomised controlled trial. To explore the application of the concept of rapid rehabilitation surgery and patient satisfaction with nursing care in the perioperative period of hip arthroplasty in the elderly.	The people in the control group received as peri-operative nursing care: orientation in preoperative exams and health education. After surgery, they were instructed to engage in early rehabilitation training according to their existing limitations to promote their recovery.	The results obtained showed that the recovery of hip function, surgery time, and hospital stay were significantly better in the experimental group than in the control group, as well as the satisfaction regarding nursing care.
[24] 242 patients divided into a control group (124) and an experimental group (118)	Randomised controlled study. To evaluate the effect of activity monitoring and text messaging on the rate of discharge to home and clinical outcomes in patients after hip or knee arthroplasty.	The control group received “usual care”. The intervention group received a physical activity monitor, daily pain tracking through text messages about post-surgery milestones and access to the doctor whenever needed. In this group, there was one branch that also received feedback with motivational messages, with goal setting and gamification. The remote monitoring and text messaging activities started at hospital discharge.	Results showed that activity monitoring and text messaging did not increase the rate of discharge home after hip and knee arthroplasty and that gamification and social support did not significantly increase physical activity. Remote monitoring demonstrated better care to people undergoing hip and knee arthroplasty by providing a direct mean of communication for the assessment of urgent problems, which translated into a significant reduction in the rate of readmissions.
[25] 144 people undergoing scheduled THA and TKA	A descriptive, retrospective, quantitative study. To analyse the gains in functional independence in patients submitted to a post-surgery motor rehabilitation programme.	The “Enable” project was created in the service, which aims to systematically promote the teaching, instruction, and training of patients who have undergone THA and TKA. In this project, the intervention of rehabilitation nursing began at home, with the correct identification of support devices, adaptive equipment, and people′s perception of their abilities. The identification of the characteristics of the home is important for the ADL training to be performed as close as possible to reality.	The results obtained in the assessment of the programme reveal an unequivocal recovery of functional independence after surgery. Thus, we can conclude that the implementation of the “Enable” project has allowed empowering the patients who underwent the post-surgery motor rehabilitation programme, THA, and TKA, as early as possible, with translation of the gains acquired in functional independence, and consequently facilitating their reintegration into society.

## Data Availability

Data are available only upon request to the authors.

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
