# Peer review of "Educational Intervention in Rehabilitation to Improve Functional Capacity after Hip Arthroplasty: A Scoping Review"

_jpm, 2022, doi:10.3390/jpm12050656_

Round 1
Reviewer 1 Report
I'd like to express my gratitude for the opportunity to review the following article on a vital subject. Kindly read my suggestions
- The abstract: the background is unclear and does not provide adequate justifications for the current study. • The abstract's stated objective differs from the one stated throughout the manuscript.
- you could complete the abstract by adding a conclusion.
- Introduction: certain sections lack references. Lines 36, 43, and 47 on page 1
- The language: I noticed that several articles were written in Portuguese. Regrettably, I was unable to verify these references because I am not fluent in Portuguese.
- Thorough English language editing is required.
- Eligibility criteria: the first paragraph is superfluous. Additionally, there was no determination regarding the methodology used in this review.
- I noticed you include articles in a variety of languages, including Portuguese, French, Spanish, and English. This is extremely interesting, but I believe you should investigate why these languages exist but not others. Consider that these articles may be overlooked by international readers unfamiliar with all of these languages.
- I disagree with the use of systematic reviews because they rely on secondary data.
- On page 6, you mention that you evaluated the strength of the evidence. Could you please provide us with additional information .
Author Response
Dear Reviewer
We would like to express our appreciation for you thorough review and the magnificent opportunity to improve the document. It was undoubtedly an opportunity to improve it, but above all an opportunity for continuous learning, for which we are very grateful.
Regarding this article, improvements to the document have been made having in account your suggestions. Hopefully we were able to clarify all the particularities mentioned.
Thank you once again for your contribution in reviewing our article.
I'd like to express my gratitude for the opportunity to review the following article on a vital subject. Kindly read my suggestions
- The abstract: the background is unclear and does not provide adequate justifications for the current study. • The abstract's stated objective differs from the one stated throughout the manuscript.
- you could complete the abstract by adding a conclusion.
We deeply appreciate the opportunity to improve the manuscript, and we have made the changed you suggested.
2. Introduction: certain sections lack references. Lines 36, 43, and 47 on page 1
We appreciate the depth of the review and have added references to the sections mentioned above.
3. The language: I noticed that several articles were written in Portuguese. Regrettably, I was unable to verify these references because I am not fluent in Portuguese.
Thank you for the suggestion, we took it into account and the articles that are bilingual we have changed the reference to the English versions (reference 1, 4, 25), although we would like to mention that the authors are Portuguese and have used some reference articles from the clinic in Portugal.
4. Thorough English language editing is required.
We appreciate the review and improvements were made, the article was written by natives and sometimes it is difficult to perfectly translate the sense of the words.
5. Eligibility criteria: the first paragraph is superfluous. Additionally, there was no determination regarding the methodology used in this review.
Thank you for the suggestion, it was taken into account and changes have been made accordingly.
6. I noticed you include articles in a variety of languages, including Portuguese, French, Spanish, and English. This is extremely interesting, but I believe you should investigate why these languages exist but not others. Consider that these articles may be overlooked by international readers unfamiliar with all of these languages.
Thank you for your comments. We appreciate deeply appreciate it. Since the authors of the manuscript are proficient in Portuguese, French, Spanish and English, it was agreed to include articles in all four languages, in a way of having a more comprehensive review.
7. I disagree with the use of systematic reviews because they rely on secondary data.
Thank you for the relevance of the review, the method we followed is in line with the JBI guidelines for scoping review which allows the inclusion of primary studies and systematic literature reviews in this type of review, whose main purpose being to map what exists on a given phenomenon. Systematic reviews following the Cochrane protocol for health interventions are more rigorous in this regard.
8. On page 6, you mention that you evaluated the strength of the evidence. Could you please provide us with additional information.
Thank you for the opportunity to improve the document, having it in account we put the reference that supports the classification
Best Regards
Reviewer 2 Report
The manuscript presents the results of a study aimed at educational intervention in rehabilitation to improve 2 functional capacity after hip arthroplasty: scoping review
The manuscript is well structured but I believe that it could be improved by taking into account the advice I have given.
Metodology
It should specify the type of instrument used to extract the data.
Result
Concerning table 3, it should be borne in mind that the main purpose of a table is the arrangement and presentation of repetitive information in an understandable form, so that large amounts of data can be arranged in a format that allows the reader to observe, study and understand them as a whole.
A good table condenses a large amount of information into a quickly understandable form and places it in a limited amount of space, saving many paragraphs of text.
In this sense I consider that it would be necessary to summarize the information presented in table 3 to the most important aspects, so that they are easy for the reader to understand.
Kind regards.
Author Response
Dear Reviewer
We would like to thank you in advance for your meticulous and detailed review. Your contribution on the review of this article was considered and improvements to the document have been made having in account your suggestions. Hopefully we were able to clarify all the particularities mentioned. The suggestions made are a useful source of knowledge that will be taken in account in future works.
Thank you once again for your contribution in reviewing our article.
The manuscript presents the results of a study aimed at educational intervention in rehabilitation to improve 2 functional capacity after hip arthroplasty: scoping review
The manuscript is well structured but I believe that it could be improved by taking into account the advice I have given.
Metodology
- It should specify the type of instrument used to extract the data.
Thank you for the review, in order to improve the document, the authors made a table to extract the data, thus allowing collaborative work.
Result
Concerning table 3, it should be borne in mind that the main purpose of a table is the arrangement and presentation of repetitive information in an understandable form, so that large amounts of data can be arranged in a format that allows the reader to observe, study and understand them as a whole.
A good table condenses a large amount of information into a quickly understandable form and places it in a limited amount of space, saving many paragraphs of text.
In this sense I consider that it would be necessary to summarize the information presented in table 3 to the most important aspects, so that they are easy for the reader to understand.
Thank you for the thorough review, we have changed the table, having removed some information, leaving only the one that answers to the research question.
Best Regards
This manuscript is a resubmission of an earlier submission. The following is a list of the peer review reports and author responses from that submission.
Round 1
Reviewer 1 Report
The authors, in this review, wanted to identify educational interventions in the rehabilitation field e identify educational interventions that improve the person's functional capacity those who have undergone hip arthroplasty for their rehabilitation process.this review is well done.